# Enhancing the power quality in radial electrical systems using optimal sizing and selective allocation of distributed generations

**Bachirou Bogno**[1,2]*, **Deli Goron**[3], **Nisso Nicodem**[1], **S. Shanmugan**[4], **Dieudonné Kidmo Kaoga**[3]*, **Kitmo**[3], **Akhlaque Ahmad Khan**[5], **Yasser Fouad**[6], **Michel Aillerie**[2]*

**1** Faculty of Science, University of Garoua, Cameroon, Garoua, Cameroon, **2** Laboratoire Matériaux Optiques, Photonique et Systèmes (LMOPS), Université de Lorraine & CentraleSupélec, Metz, France, **3** Department of Renewable Energy, University of Maroua, National Advanced School of Engineering of Maroua, Maroua, Cameroon, **4** Department of Engineering Physics, Research Centre for Solar Energy, College of Engineering, Koneru Lakshmaiah Education Foundation, Green Fields, Vaddeswaram, Guntur, Andhra Pradesh, India, **5** Department of Electrical Engineering, Integral University, Lucknow, India, **6** Department of Applied Mechanical Engineering, College of Applied Engineering, Muzahimiyah Branch, King Saud University, Riyadh, Saudi Arabia

* kidmokaoga@gmail.com (DKK); boogno@gmail.com (BB); michel.aillerie@univ-lorraine.fr (MA)

**Data Availability Statement:** All relevant data are within the manuscript and its Supporting Information files.

## Abstract

Optimizing energy resources is a major priority these days. Increasing household energy demand often leads to the deterioration of poorly sized distribution networks. This paper presents a method for energy compensation and optimization in radial distribution systems (ORDS). By integrating distributed generations (DG), an approach is used to evaluate voltage and power profiles, as well as power losses on radial distributed systems (PLRDSs). After integrating distributed generations, improved voltage and power profiles are established. A potential solution to power compensation and blackouts (PCB) can also be the use of hybrid distributed generation systems (HDGSs) that reinforce radial distribution networks (RDNs) by improving power quality. Accordingly, a proposed configuration system is shown in this work to inject multiple renewable energy sources (MRES) from selected regulated nodes. The feasibility of the proposed system is evaluated using particle swarm optimization (PSO), which was used to locate stable nodes and locations, sensitive to voltage fluctuations. The proposed approach is based on the evaluation of the power losses and voltage profiles of the IEEE 33 bus and IEEE 69 bus standards This MATLAB-based method establishes an objective function that converges more quickly to the optimal results.

## 1. Introduction

### 1.1.Challenges of recent electricity market

Power outages have worsened as a result of falling oil prices caused by conflicts in the Middle East and Ukraine. The widespread consumption of electricity has become an urgent problem for nations worldwide [1]. Nonetheless, fossil fuels account for a significant portion of electrical power usage, accounting for over 75% of the total electricity use when compared to other energy sources [2]. However, the rise in greenhouse gas emissions (GHGs), which cause

**Funding:** The authors extend their appreciation to the Reserachers Supporting Project number (RSPD2025R698), King Saud University, Riyadh, Saudi Arabia for funding this research work.

**Competing interests:** The authors have declared that no competing interests exist.

climate change, is due to the consumption of these fossil fuels [3]. Making a dependable energy plan that can use sustainable, non-polluting energies is crucial if we want to reduce our reliance on fossil fuels [4]. Because of this, power generation from renewable sources has continued to be a major concern for a great number of nations for the past decade [5]. Integrating those sources into the electrical networks, however, presents a different challenge [6]. Power losses and voltage level instability during links create a serious issue for power distribution enterprises.

For this reason, there are a number of guidelines for the integration of primary energy sources into power systems [7]. Also, these organizations' operational expenses are going up and their revenues are going down because of these massive problems [8]. For an extended period, most organizations have heavily incorporated renewable energies into distribution systems, focusing on ensuring reliable generating systems, regulating voltage levels, and managing active and reactive power issues [9]. The disruption of management and control systems is another potential outcome [10]. If we want distribution systems to be secure, stable, and efficient, we need to make sure that the distributed generation (DG) units follow all applicable laws and regulations [11]. Consequently, modern distribution network (DN) management presents a number of obstacles. Maintaining a consistent and controlled voltage profile is the primary goal of distribution networks [12]. To provide uninterrupted service over the anticipated operational term, electrical equipment requires a quality voltage supply, which is of utmost significance to consumers [13].

Multiple studies have identified DGs as a way to meet these objectives by increasing load voltages and power consumption. The goal of installing these DGs at each cable feeder's terminus is to improve voltage profiles and reduce power losses. The DGs appear to improve power and voltage patterns, distribution network dependability, and energy flow. For distribution networks, reliability is paramount. A more consistent and dependable power supply is one benefit of incorporating DGs into electrical distribution networks [14]. Not only that, it's cheap. The majority of benefits offered to utilities and consumers include decreased energy loss and improved voltage patterns [15]. Other benefits include lower transient conditions and peak load shaving, which facilitate DGs' connection to distribution networks.

Multiple DGs in a network can create a number of technical issues: overvoltage, undervoltage, overloads, system protection faults, harmonics that affect power quality, and overheating of distribution line feeders [16]. The most frequently used DGs include wind power plants, biomass and photovoltaic systems. These sources are the main basic technologies for distributed generation. Several countries are encouraging decentralized production (DP) due to very high electricity costs. This has led to an increase in DG technology as a major potential in the electricity market (ELEM). The total quantity of DGs in the US, for instance, rose from 2003 to 2023 [17].

Advanced knowledge of production and management approaches is required for DG technologies. DGs employ a variety of approaches and assessments to evaluate the conformance and the feasibility of the data they generate. Furthermore, DGs can use optimization techniques to manage and control their energy flow, allowing for the analysis of power losses, voltage profiles, and DG scale [18]. Modern algorithms, such as meta-heuristic procedures, provide the possibility of systems optimizations.

## 1.2. State of the art on the challenges of distributed and decentralized generation

Methods for incorporating DGs into distribution networks constitute the current energy market's highest priority [19]. This method's stated goals are to reduce transient times caused by harmonics, improve steady-state system stability, reduce power losses caused by disturbances,

decrease line current losses, and improve load and network voltage profiles [20]. This led to the creation of numerous approaches and algorithms aimed at optimizing and integrating DGs into distribution networks. The goal of these approaches is to regulate DG usage. Radial distribution networks employ multiple approaches for DG placement. For the most part, these approaches depend on heuristics and metaheuristic optimization approaches [18]. These procedures, which draw inspiration from biological or natural phenomena, determine placement and size distribution. Although genetic algorithms (GAs) are advanced techniques, the whale optimization techniques, and PSO are among the most widely used algorithms [21]. However, many studies point to the fact that hybrid PSO algorithms provide better optimal placement of STATCOMs [22]. There is also multi-objective optimization, the evolution is based on Pareto methods, salp swarm algorithms and ant-lion optimization. Some methods use smart inverters to control the voltage in a radial distribution system, which also allows good voltage stability at the level of the system buses, in order to correct the power factor [23]. DGs have the advantage of solving voltage instability problems when properly dimensioned [24]. Their high degree of penetration in systems is a major advantage. However, DGs are less dependable than other power sources due to their intermittency, which may cause significant disruptions to energy output. In [25], a method for planning and controlling grid-connected and off-grid systems is also proposed. This method has made it possible to allocate surplus energy from solar farm systems to other, non-proprietary operators. In [26], new approaches for allocating the size and location of DGs in stand-alone microgrids are proposed using system reconfiguration according to some topologies; for this method, power losses were very high in the distribution network although optimal network reconfiguration (ONR) and improvement of voltage profiles in electrical distribution networks (EDNs) were proposed. In [27], other researchers studied and proposed the implementation of new algorithms to solve power loss problems in DGs, but voltage profiles were not taken as an important parameter since this was applied to an asynchronous machine; their goal was to reduce real power losses. However, in [28], a method of optimizing energy efficiency using the integration of DGs was proposed, which made it possible to increase the voltage profile of the distribution network, even though the loads were resistive. They used harmony search algorithms, genetic algorithms, and algorithms consisting of PSO hybridization. Achieving the best possible result after system reconfiguration is the goal of the techniques offered in [17], which use ant colony optimization and the firefly algorithm for network reconfiguration. In [18], it is demonstrated how a radial system with numerous DGs improves the network's performance. This technology allows you to change the bus voltage amplitude, limit line current loss, reduce voltage deviation, and minimize power loss. Regrettably, without taking into account voltage restrictions, we cannot guarantee voltage stability or reduce system power losses[29]. When studying and evaluating the effects of DGs on distribution networks, we must consider harmonic limits, voltage profiles, deviations from those profiles, power losses, and line current losses. In this paper, bus and node testing on IEEE 33 and Improved IEEE 69 standard is used to enable validation and reliability of the proposed method. The required data are modelled and analyzed using MATLAB/Simulink software.

Our study has established the best method for estimating the total system configuration size. We also assess power loss using IEEE 69 and IEEE 33 bus testing, as well as identifying the allocation nodes for the different DGs.

## 2. Power management due to load demand

Using this method of energy management (EM), limitations related to quality of service and demand are considered. We therefore have the objective function described in **Eq (1)** and the

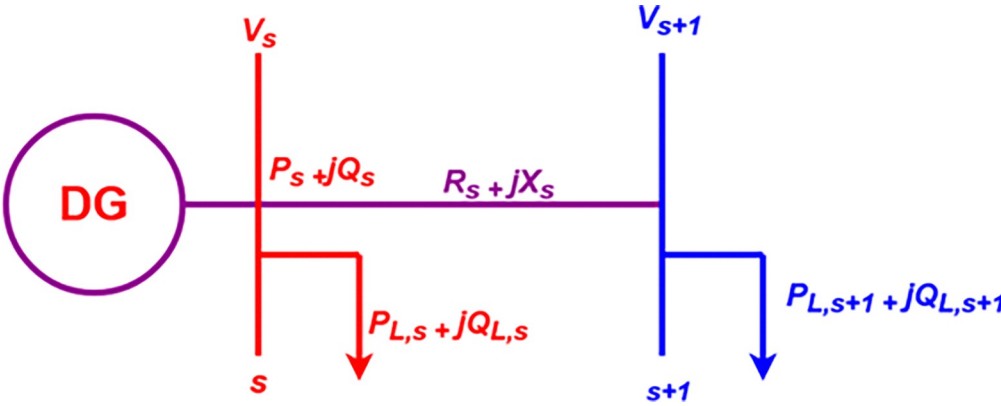

**Fig 1. Segment of the radial system linking two nodes.**

constraints defined in **Eq (2)**. $P_{ls}$ represent the power losses due to the disturbances on overall system using different sources of power generation. The $P_{Load}$ is the power the load. **Fig 1** illustrates the first step of modelling the buses and nodes in MATLAB as in **Eq (1)**. This is a portion between two nodes of the radial electrical system.

In the current configuration, $P_{ls}$ represents the power losses [30], $R_s$ the line resistance, *lo* the active power and $Q_s$, the reactive power, and $V_s$ the k portion voltage. The same is valid for the nominal currents of the input and output voltages of the bus which are respectively: $I_s$, $I_{o,s}$, $V_i$ and $V_o$. Also, *lo* represents the length of a radial system configuration branch. $P_{Syst}$ is the power of overall system, $P_V$ the power of photovoltaic plant, $P_W$ the power of wind farm, $P_{Grid}$ the power of grid side, and $P_{Load}$ the power of load demand.

$$P_{ls} = \sum_{s=1}^{lo} R_s \times \left( \frac{P_s^2 + Q_s^2}{V_s^2} \right) \tag{1}$$

$$constraints \begin{cases} P_{Syst} = P_V + P_W + P_{Grid} \\ P_{Syst} \succ P_{Load} \\ P_{Load} \succ P_{ls} \end{cases} \tag{2}$$

$$V_i \leq V_s \leq V_o; \text{k} = 1, 2, \ldots, \text{lo} \tag{3}$$

$$0 \leq I_s \leq I_{o,s}; \text{s} = 1, 2, \ldots, \text{lo} \tag{4}$$

In order to model the different energy sources of distributed generators, the $P_{dg\_s}$ and $P_{dg\_o,s}$ powers are used to model the required power supplied by a line of length xo. The voltage limits are defined by **Eqs (3)** and **(4)**.

$$0 \leq P_{dg\_s} \leq P_{dg\_o,s}; \text{s} = 1, 2, \ldots, \text{xo} \tag{5}$$

## 3. Objective function

The objective function is defined with the aim of reducing power losses on the one hand and reducing the voltage difference between adjacent nodes on the other. The aim of network reconfiguration and distributed generation allocation is to balance the distribution systems.

**Eq (9)** defines this objective function. In this formula, the parameters Ps, Qs, Rs and Vs represent respectively: the active power at node s, the reactive power at node s, the resistance at node s and the voltage amplitude of a branch number s. $\Delta \hbar$ is the voltage stability index, $\Delta P_{ls}^{lo}$ is the power stability index.

$$\Delta P_{ls}^{lo} = \frac{P_{ls}^{lo}}{P_{ls}^{o}} \tag{6}$$

$$\hbar_{s+1} = |V_s|^4 - 4(P_{s+1}X_s - Q_{s+1}R_s)^2 - 4(P_{s+1}R_s - Q_{s+1}X_s)|V_s|^2 \tag{7}$$

$$\Delta \hbar = \max\left(\frac{1 - \hbar_s}{1}\right); \quad s = 2, \dots, l_o \tag{8}$$

$$\min(Fitness) = \Delta P_{ls}^{lo} + \Delta \hbar \tag{9}$$

## 4. Evaluation using distributed generation approaches

### 4.1. Test using 33 nodes standard system

The architecture and configuration of an IEEE standard test system [31] is important in addressing the various constraints for a radial distribution system. **Fig 2** displays the IEEE 33 bus. This picture shows three options for adding additional dispersed-generating sources. These are nodes 13, 17 and 25. This is possible after reconfiguring the system, which consists of three distributed generations: DG1, DG2 and DG3. The sources are defined by the photovoltaic system for DG1, the wind plant for DG2 and the electricity grid for DG3.

### 4.2. Test using 69 nodes standard system

The system configuration for a 69 bus system [31] is shown in **Fig 3**. Reconfiguring the system in this way enables testing of power loss and system sizing. This technology clearly has the

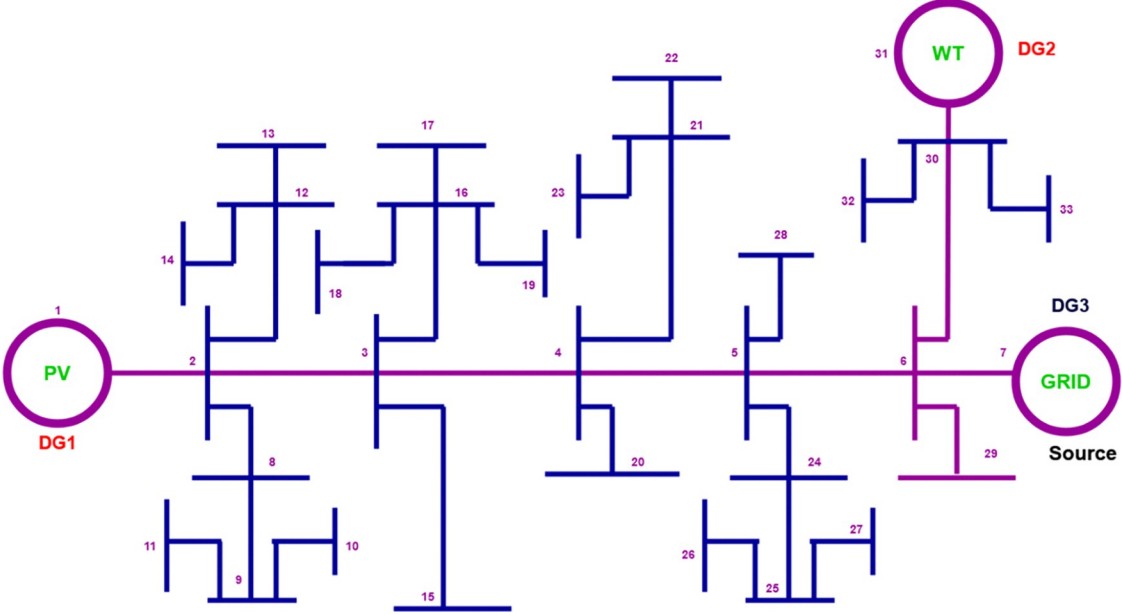

**Fig 2. IEEE 33 nodes configuration on radial electrical system.**

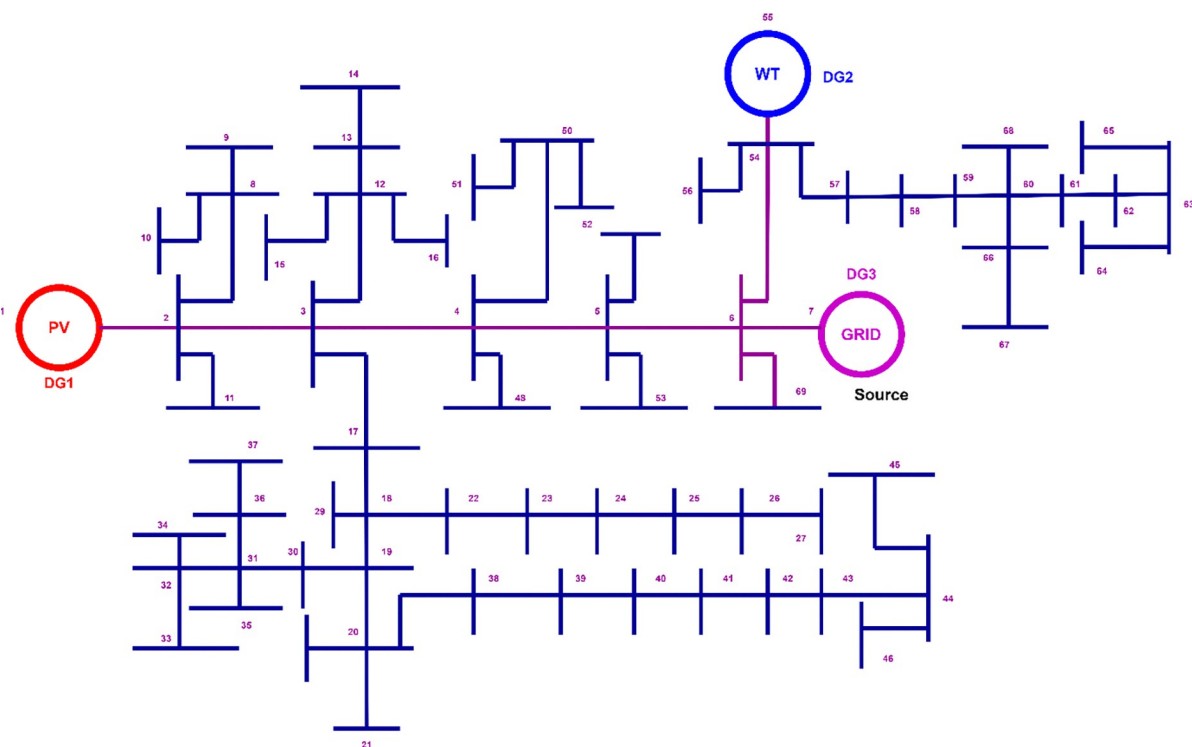

**Fig 3. IEEE 69 nodes configuration on radial electrical system.**

capability to accommodate many DG installations. In particular on buses 9, 14, 44, 60 and 63. The nodes at these points are steady in current and voltage for a power profile whose power losses are minimized.

## 5. Problem formulation

In order to minimise voltage drops and maximise energy losses, this study aims to assess and demonstrate the practicability of optimally integrating resources based on distributed generation in systems linked to electrical grids. We suggest PSO techniques because this optimization issue involves multiple agents and multiple objectives. The distribution network under consideration has a radial architecture and the R/X ratio is emphasized for maximum minimization of power losses and reduction of voltage dips on the system. It has been proven that the Newton-Raphson method regarding load flows is best suitable for solving the problems of systems involving distributed generations. This method calculates the load flow on the various nodes of a distribution network by considering the bus voltages, the line currents and the real and reactive power losses on each bus feeder. This article deals with sizing and determining the injection locations for renewable energy sources. The aim of this DG placement is to improve the voltage profile and minimize the power losses over the whole base distribution system which is supposed to be strengthened and reconfigured. Limitations on obtaining the most realistic and accurate voltage profile and power loss estimates are among the optimization criteria taken into account. Additionally, a portion of line located between $s$ and $s + 1$ locations, an impedance of $R_s + jX_s$ and the voltage drops at bus s and s+ 1 are considered in **Fig 1** above. This figure is an illustration. $P_s$ and $Q_s$ are the bus real and reactive power flows. At nodes $s$ to $s + 1$, we have $V_s$ and $V_{s + 1}$ respectively, the complex voltages. The power losses between buses $s$ and $s + 1$ are evaluated and represented by $P_{ls}$ according to **Eq (1)**.

## 6. Optimization approach

In [32], the authors present an overview of the literature on methods using PSO algorithms. Recent studies have focused on the optimization, placement and sizing of DGs in distribution networks. In [33], a review of these methods summarizes several algorithms that are suitable for sizing distributed generations [34]. We use this approach to identify the optimal solution among a set of distinct challenges. In addition, it is necessary to increase the number of iterations in order to improve the convergence of the problem for better performance. We refer to each particle as an "individual." We modify them at every step and iteration, following the steps outlined in **Fig 4's** flow diagram. This investigation establishes an objective function in order to achieve superior global exploration fields ($g_{best\alpha}$). This investigation establishes an objective function, adhering to the limitations in **Eqs ([3]) and ([4]).**

A random generation of particles in the solution exploration space is always carried out. The current positions and velocities of these particles are denoted by $X_\alpha^\delta$ and $V_\alpha^\delta$ respectively. The algorithm must allow each individual particle $\alpha$ to find the best personal positions in the chosen search field as given in **Eqs ([10])** and **([11])**.

$$X_\alpha = (x_{\alpha,1}, x_{\alpha,2}, \ldots, x_{\alpha,f}) \tag{10}$$

$$V_\alpha = (v_{\alpha,1}, v_{\alpha,2}, \ldots, v_{\alpha,f}) \tag{11}$$

For fast convergence speed to the best results, six optimal PSO parameters need to be considered: $c1$, $c2$, $\omega_o$, $\omega_i$, $\hbar_i$ and $\hbar_o$. These are efficiency and reliability parameters of PSO algorithms. With these parameters in mind, values are set such that $c_1 = c_2 = 2.1$. These are the values of the individual and social acceleration coefficients. The values of the weights or coefficient of inertia are given as $\hbar_o = 0.9$ and $\omega_i = 0.5$ corresponding to the particle velocities. $\hbar_o$ represents the maximum number of iterations, number of particles by $\hbar_i$. For good robustness, the value of $\hbar_i = \hbar_o$ are such that $\omega_\alpha < \varepsilon$, where $\varepsilon$ represents the minimum error. The flowchart of the PSO algorithm is given in as in **Fig 4**.

$$P_{best\_\alpha} = (p_{best\_\alpha,1}, p_{best\_\alpha,2}, \ldots, p_{best\_\alpha,f}) \tag{12}$$

$$G_{best\_\alpha} = (g_{best\_\alpha,1}, g_{best\_\alpha,2}, \ldots, g_{best\_\alpha,f}) \tag{13}$$

The swarm particles must be updated to have the best positions corresponding to $P_{best\_\alpha}$ and $g_{best\alpha}$ which are assigned to the velocity vectors according to **Eqs ([12])** and **([13])**, and the current positions $V_\alpha^\delta$. The parameters $c_1$ and $c_2$ refer to the capacity with which a PSO particle moves towards *Pbest* and *gbest* with velocities $V_\alpha^\delta$.

$$V_\alpha^{\delta+1} = \omega_\alpha V_\alpha^\delta + c_1 r_1 (P_{best\alpha} - X_\alpha^\delta) + c_2 r_2 (g_{best\alpha} - X_\alpha^\delta) \tag{14}$$

$$X_\alpha^{\delta+1} = X_\alpha^\delta + \phi(v_\alpha^{\delta+1}); \qquad \delta = 1, 2, \ldots, f \tag{15}$$

In **Eq ([16])**, $\phi$ is the constriction factor, $\phi$ is the iterative number. The inertia weights govern the convergence performance of the PSO.

$$\omega_\alpha = \omega_o - \frac{(\omega_o - \omega_i)}{\hbar_o} \hbar_i \tag{16}$$

By using PSO techniques, it is possible to achieve regular and rapid convergence towards the best scores. The overall search capacity to obtain $P_{best\_\alpha}$ and $g_{best\alpha}$ depends on the efficiency of the algorithm chosen and the robustness of the system.

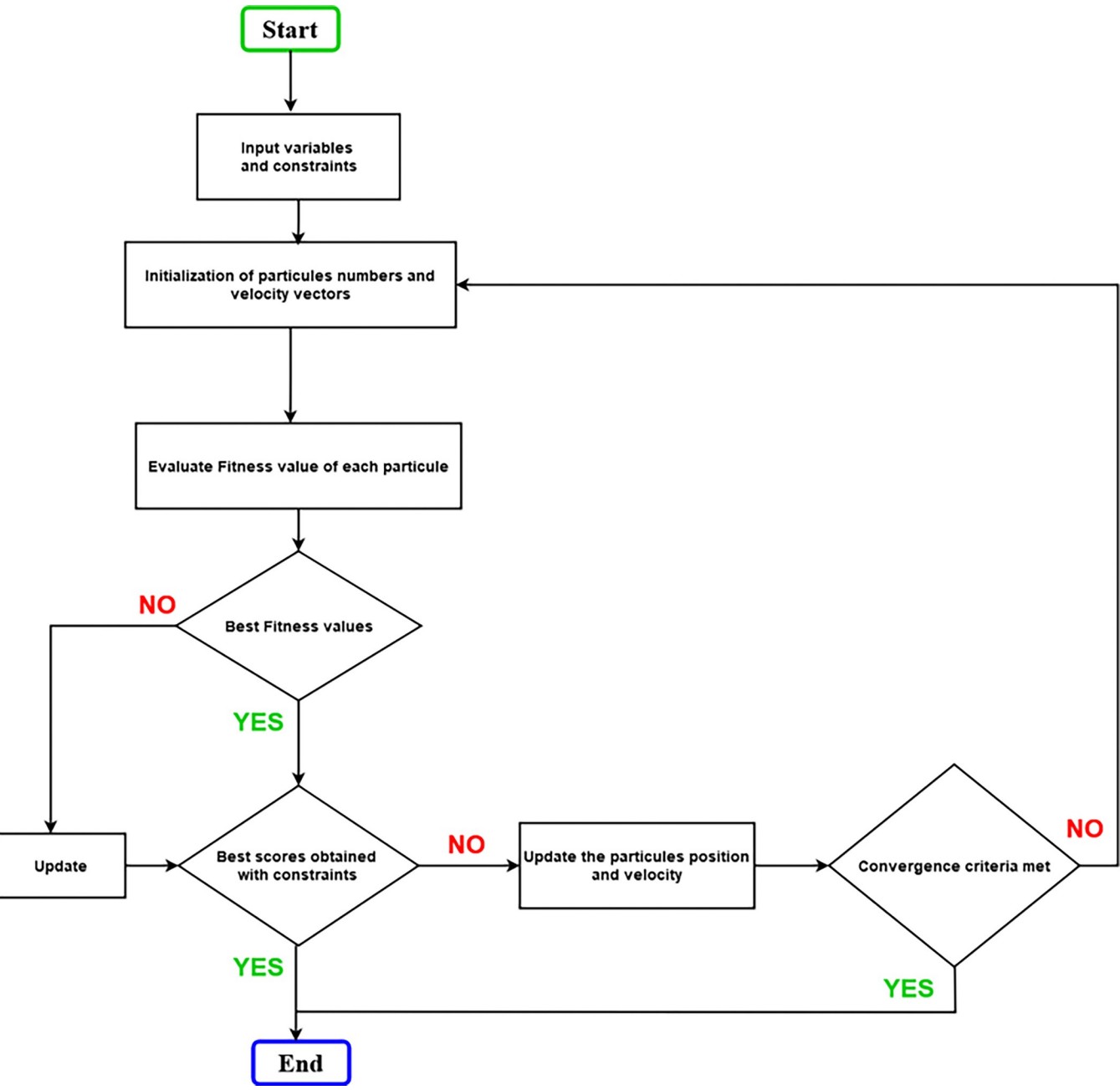

**Fig 4. Process on flowchart of proposed technique.**

## 7. Results and discussion

### 7.1.Evaluation of the connection to the electrical grid using IEEE standard

A considerable development has focused on the regulation [35] and stability [36] of voltages in electrical distribution systems. In the same context, this work aims to improve the voltage and power profile based on standard tests [37]. **Fig 5**, which illustrates the system voltage profile for the combination of three approaches, shows the outcomes of using several algorithms to determine the system's resilience. By combining the PSO and genetic algorithms, we improve

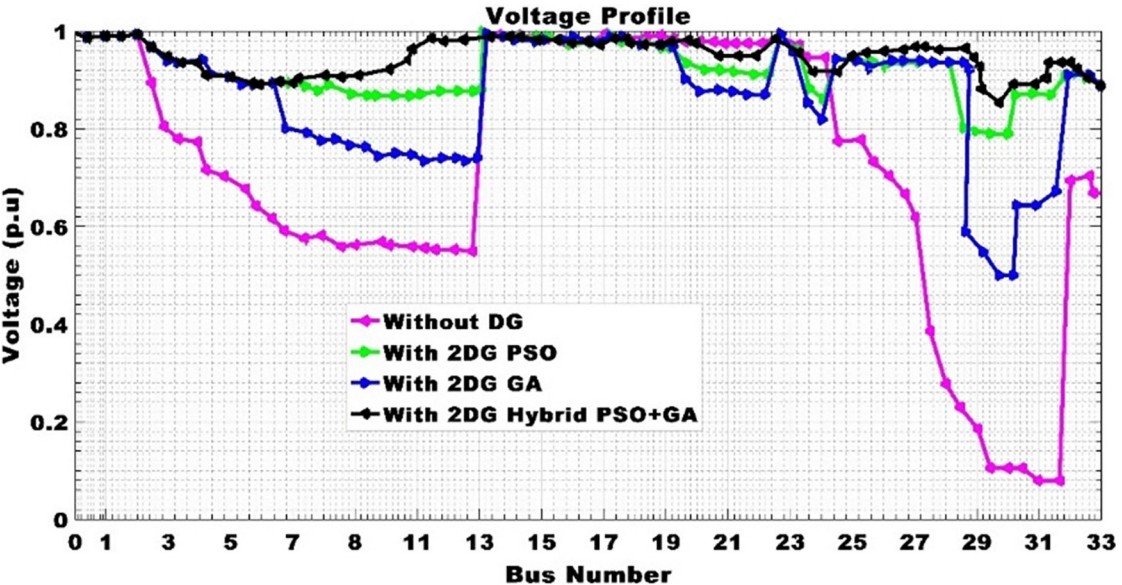

**Fig 5. Scenarios for algorithms using IEEE 33 bus.**

the voltage profile instead of using single approach. Simultaneously, the system scale and the number of stability nodes are calculated. Buses 15, 17, 23 and 27 all exhibit voltage stability during the system's steady state. These instants correspond to the periods during which the loads in the households can receive stable voltage levels, present at the relative buses. On the other hand, buses 5, 7, 8, 11, 29 and 31 are unreliable for supplying loads, because the voltage profiles are unstable. The injection of a source or the supply of a load from a bus or node is conditioned by the stability at these locations, which makes it possible to choose the types of loads to be supplied. On the other hand, unstable voltage levels make it necessary to add numerous production sources to the radial system to make it stronger or more stable.

Fig 6 also presents voltage profiles. We identified these voltage profiles by combining several algorithms: cuckoo search, GA, PSO, and the Whale Optimization Algorithms (WOA). The PSO algorithms performed exceptionally well. Combining two dispersed generations makes the voltage on buses 30, 35, 40, and 45 more stable. Buses 15, 20, 61, and 65 experience system instability when the two sources are not integrated. Compared to the other related algorithms, the PSO algorithms clearly show superior performance. Even when using multiple sources, the system maintains stability due to its fast convergence time and improved outcomes.

## 7.2. Allocation and sizing of the various generations distributed

In this study, the radial distribution network was chosen to be studied on the IEEE 69 and IEEE 33 systems. Figs 7 and 8 depict the power profiles from testing on the standard IEEE 69 bus and the standard IEEE 33 bus, respectively. With these buses, we can test different scenarios and determine how much system size can be allocated on the networks under consideration. This method is applied to accurately know the quality of the primary energy sources required to be injected at a node in radially configured systems. Integration scenarios are proposed for several possible configurations in this work; for example, using a single source in the grid results in excessive power loss, but the PV+wind (2DG) scenario, which combines the grid with a battery, improves the power profile. In addition, the voltage at each node on buses

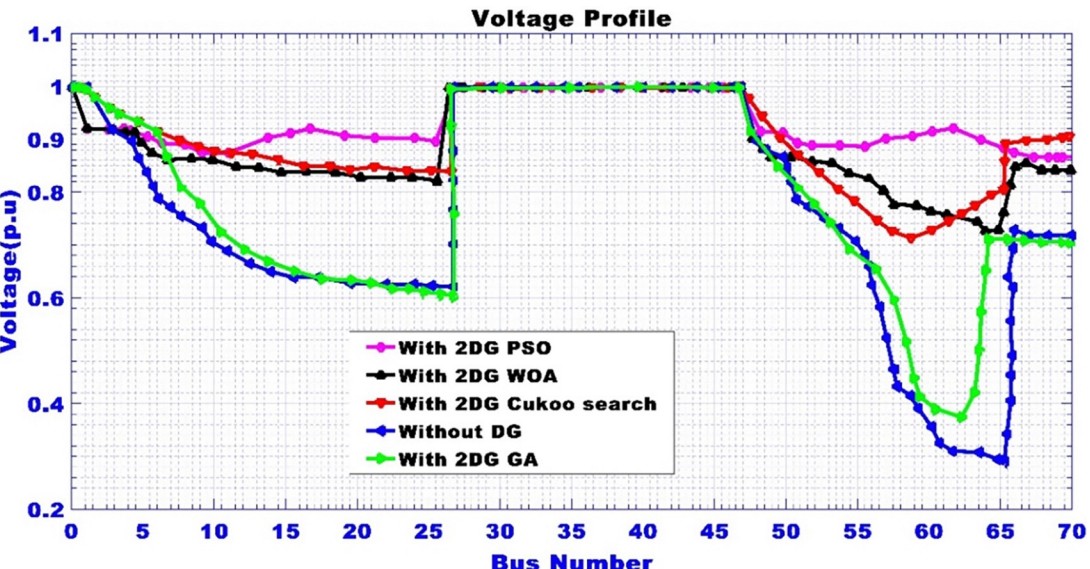

**Fig 6. Scenarios for algorithms using IEEE 69 bus.**

33 and 69 remains constant for this scenario. Under this configuration, different sizes of generation sources can be injected and combined, thanks to the main sources of the hybrid system, which include photovoltaic and wind power. The 33-bus radial network shows stability at nodes 14, 15, 16 and 17, but disturbances are observed at nodes 6, 7, 8, 29 and 32 of bus 33. On the other hand, Fig 8 shows the voltage profile for the grid + battery + PV + wind scenario. Nodes 30, 31 and 33 of the 69 buses remain within the voltage and power stability margin, while nodes 15, 20, 21, 23, 62, 63 and 65 show instability.

In Fig 7, without the DG (magenta colour), the voltage drop is high. With 1DG (blue colour) the voltage decrease improves. When 2DGs (green colour) and 3 DGs (black colour) are

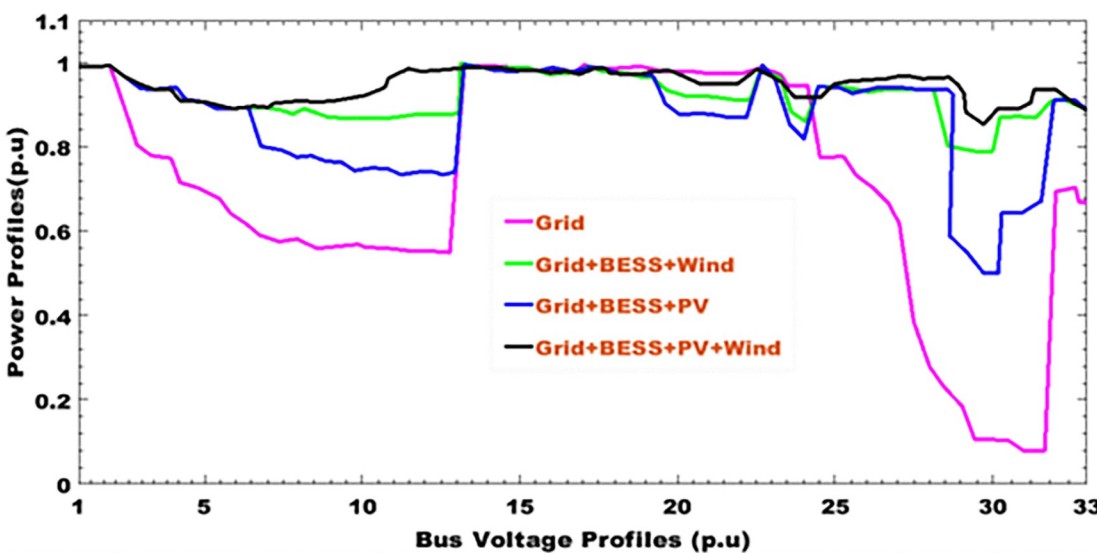

**Fig 7. Power profiles for scenarios using IEEE 33 bus.**

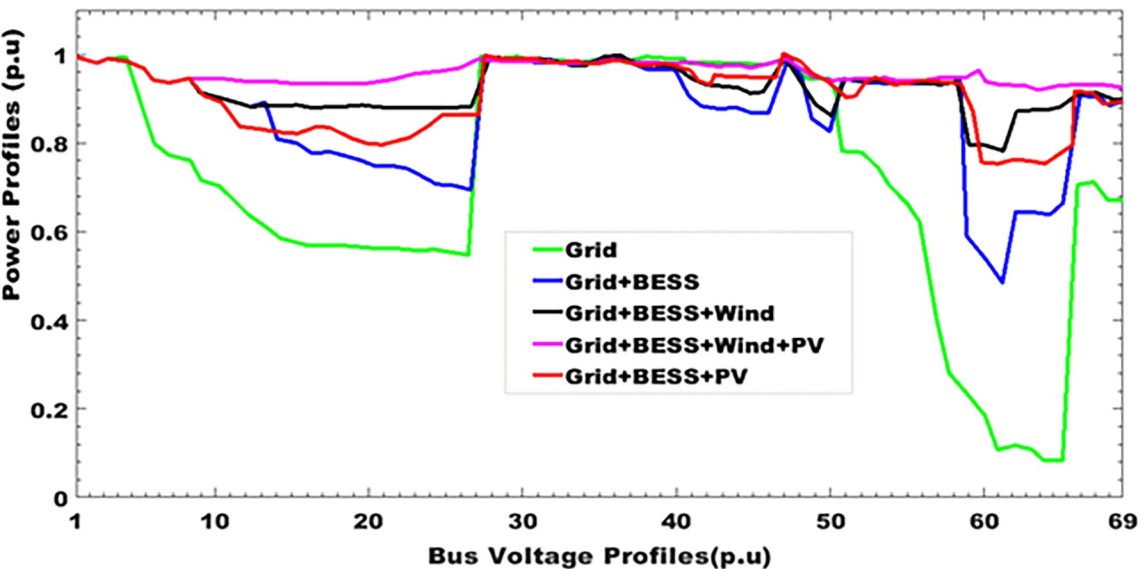

**Fig 8. Power profiles for scenarios using IEEE 69 bus.**

installed, the voltage profile improves and the system becomes stable. The same applies to **Fig 9**: with 1DG (magenta colour) or (blue colour), the power losses are very significant; when 2DGs (black colour) or (red colour) are injected, the power losses progressively decrease.

### 7.3. Evaluation of the power losses IEEE standard test

In **Fig 9,** we can see the power loss attributes at each node in the radial network. Using the IEEE 33 bus standard test, we observe that buses 3 and 7 possess significant power losses of 56 kW and 40 kW, respectively. Buses 1 and 27 lose an estimated 10 kW of energy. Furthermore, the wind farm itself is responsible for these power losses. Integrating the wind farm, PV plant,

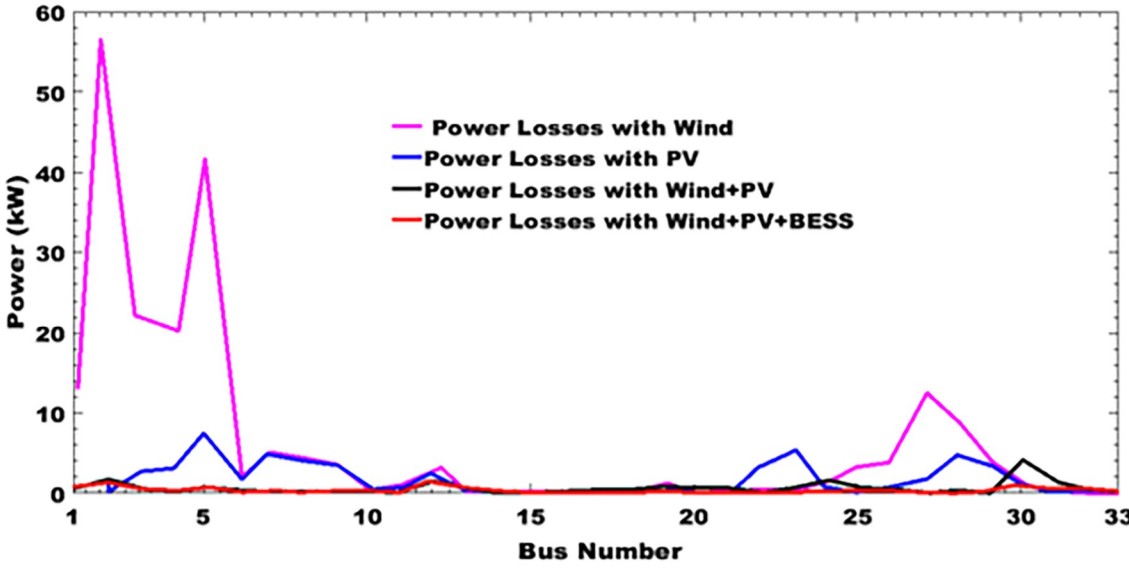

**Fig 9. Power loss profiles using IEEE 33 bus.**

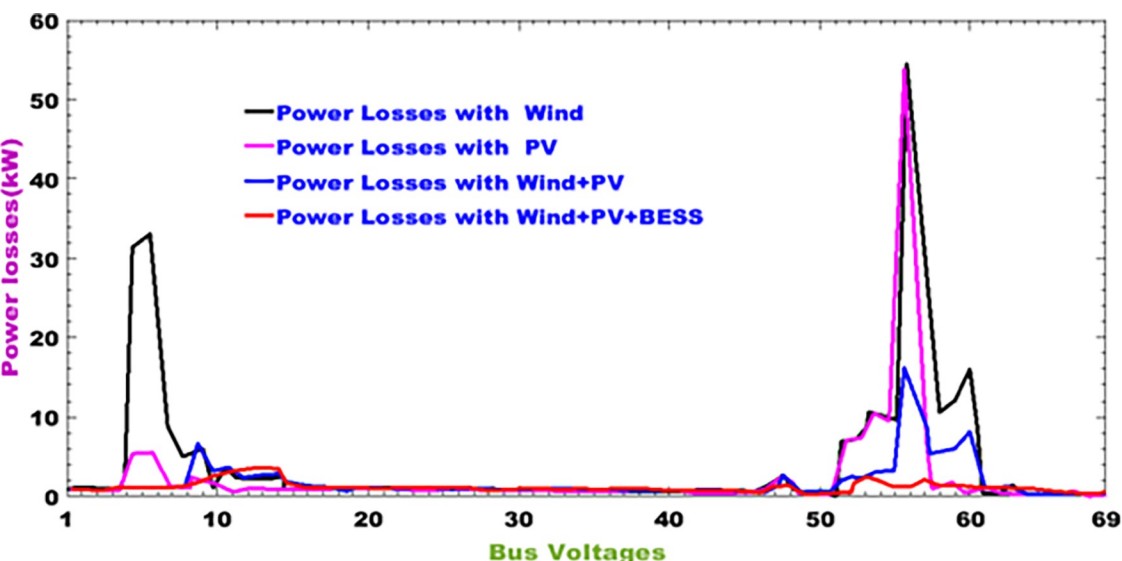

**Fig 10. Power loss profiles using IEEE 69 bus.**

and battery banks into the grid significantly lowers power losses, reaching a maximum of around 5 kW.

**Fig 10** shows other bus layouts that adhere to the IEEE 69 standard; bus 8 has power losses of 31 kW, whereas bus 55 experiences losses of 51 kW. Combining the PV wind farm and battery storage technology significantly reduces power losses to 4 kW. Power losses on bus 69 are quite small in comparison to bus 33 test standard.

The system is reconfigured by opening some switches, as demonstrated in **Figs 11** and **12**, and by closing the switches involved in switching. The system ensures automatic

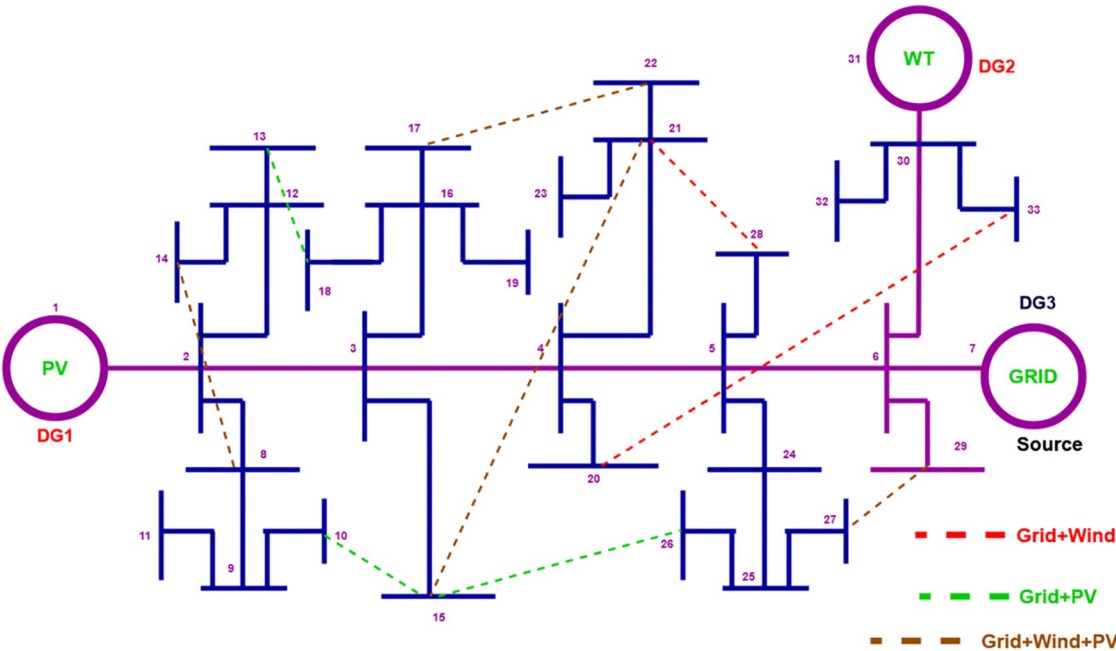

**Fig 11. Topology of the system after reconfiguration on IEEE 33 bus.**

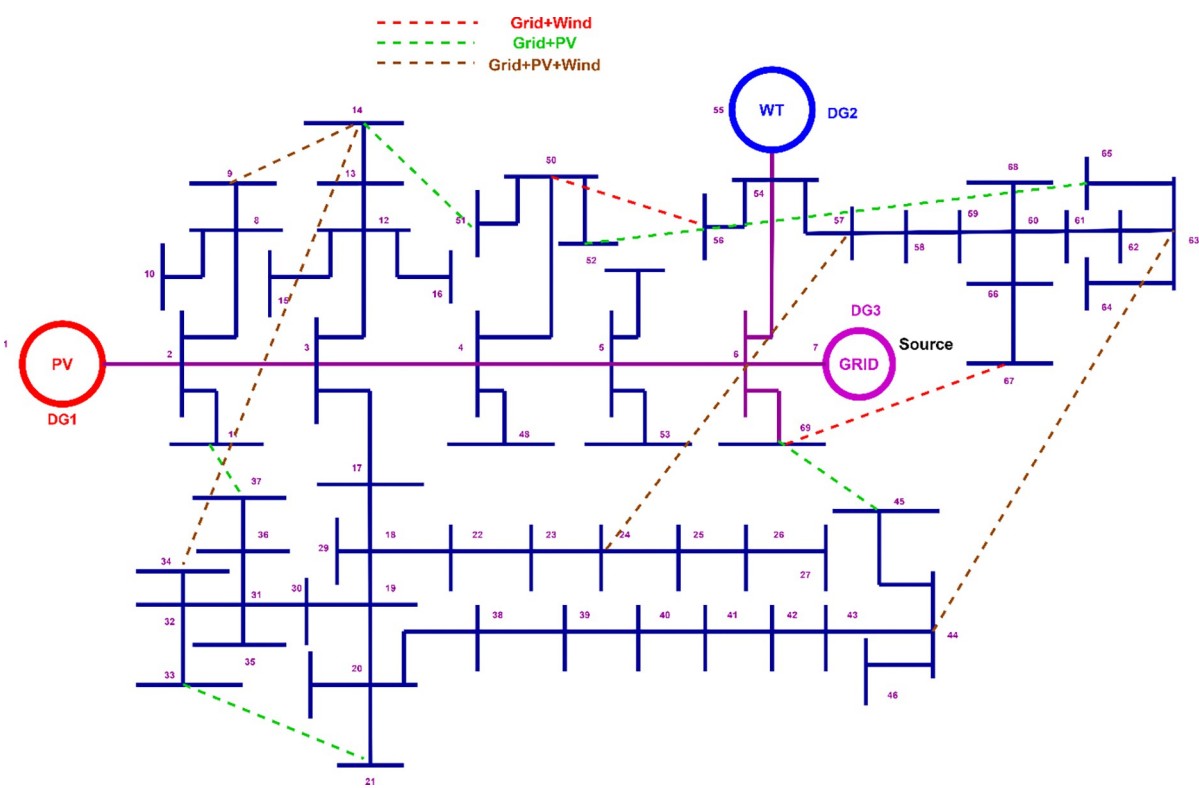

**Fig 12. Topology of the system after reconfiguration on IEEE 69 bus.**

reconfiguration to reduce power losses and to meet the energy demand by the load at the common point of coupling. Energy management and system reconfiguration depend on the power required by the loads. In this case, opening or closing allows the system's active and reactive power to be increased or reduced. Three scenarios are presented: Grid+Wind (Scenario 1), Grid+PV (Scenario 2) and Grid+Wind+PV (Scenario 3). The maximum power level is reached with the Grid+Wind+PV scenario. In all three cases, the branches are connected and disconnected to create other meshes and loops.

Initially, some switches are open, while others are closed, both for the IEEE 33 bus standard configuration (**Figs 2** and **11**) and for the IEEE 69 bus standard configuration (**Figs 3** and **12**). **Table 1** shows the switch positions. Opening or closing is managed by PSO algorithms, which take account of blackouts and sudden and unexpected changes in the behavior of non-linear loads.

## 7.4.Optimization of the hybrid system using meta heuristic algorithms

In **Fig 11**, four methods are used to optimize the system as a whole. We use the different algorithms to establish the sizes of the different DGs and distribute them in the radial design

**Table 1. State of switches for three scenarios.**

| Scenarios | IEEE 33 Bus | IEEE 69 Bus |
|---|---|---|
| Opened switches before configuration | 15-16-23-30 | 5-15-21-41-53-56 |
| closed switches after configuration | 3-5-12-16 | 17-20-25-27-42-54-58-62-65 |
| Possibility of DG units allocation | 8-15-21-25-27-29-30 | 13-16-20-21-24-33-37-41-45-52-54-60-61-67 |

Table 2. *Evaluation of the hybrid system for IEEE 33 bus.*

| nBus | System size (MW) | Loss (kW) | Loss (kVar) | nBus | System size (MW) | Loss (kW) | Loss (kVar) |
|---|---|---|---|---|---|---|---|
| 1 | 8.268 | 8.780 | 21.440 | 18 | 5.930 | 4.258 | 10.728 |
| 2 | 8.068 | 10.580 | 23.760 | 19 | 5.276 | 4.788 | 10.882 |
| 3 | 8.344 | 7.198 | 16.569 | 20 | 4.188 | 5.701 | 13.300 |
| 4 | 4.531 | 9.117 | 18.691 | 21 | 4.434 | 6.939 | 14.584 |
| 5 | 7.300 | 5.661 | 11.894 | 22 | 3.214 | 7.594 | 16.576 |
| 6 | 7.522 | 5.662 | 12.020 | 23 | 3.127 | 7.705 | 16.838 |
| 7 | 7.500 | 4.659 | 10.897 | 24 | 5.783 | 5.687 | 11.739 |
| 8 | 6.7758 | 5.157 | 11.930 | 25 | 4.968 | 6.990 | 14.698 |
| 9 | 6.996 | 5.243 | 12.208 | 26 | 3.110 | 7.178 | 17.749 |
| 10 | 5.028 | 6.993 | 13.124 | 27 | 5.773 | 6.993 | 13.124 |
| 11 | 5.693 | 6.192 | 13.618 | 28 | 6.486 | 6.192 | 13.618 |
| 12 | 5.600 | 6.339 | 13.948 | 29 | 6.565 | 6.339 | 13.948 |
| 13 | 5.773 | 6.771 | 14.206 | 30 | 6.195 | 6.771 | 14.206 |
| 14 | 6.486 | 4.639 | 10.018 | 31 | 6.535 | 4.639 | 10.018 |
| 15 | 6.565 | 4.432 | 10.937 | 32 | 5.930 | 4.432 | 10.937 |
| 16 | 6.195 | 4.710 | 10.880 | 33 | 5.276 | 4.710 | 10.880 |
| 17 | 6.535 | 4.367 | 10.430 | | | | |

system: PSO, Cuckoo Search and WOA. We found optimal convergence speeds with n = 3.6 for PSO, n = 5.55 for Cuckoo Search and n = 20.51 for WOA. Clearly, PSO provides a robust evaluation and optimization of the whole system. These algorithms use the input and output characteristics of the system to evaluate and size the system.

The total size of the whole system is determined after the losses have been assessed. It is also important to specify and locate the voltage level and power size to be allocated to each node in the radial system. **Table 2** shows the results of the IEEE 33 test, which determines the different voltage levels throughout the system. It is clear that the average power supplied by the system overall is 6.195 MW. An estimated 5,185 kW of power is lost. However, the reactive power losses are 10.124 kVAR. Considerations such as system design and the expected number of nodes can be known using the IEEE 33 test. To assess daily efficiency, we examine these losses for 24 hours once a year. The key factors in DG sizing are minimizing power losses and improving voltage profiles as defined in Table 2. This table summarizes the different voltage and power levels obtained after automatic reconfiguration of the system. It can be seen that by adding DGs, the voltage profile improves and power losses also decrease.

The PSO algorithms are compared to two other algorithms in **Fig 13**. The speed of convergence of the objective function shows that PSO converges towards the best scores compared with the other two algorithms. These results are in the same direction as those obtained in [19]. The aim of optimizing the system is to reduce power losses and improve the voltage profile on the various branches and nodes of the radial system. The algorithms also make it possible to reduce the transient interval in order to limit power losses. The work in [20] shows that PSOs are suitable for reducing power loss and are better suited to the constraints associated with the fast response of an electrical system. **Table 3** shows the results obtained by the algorithms selected from the literature [38]. Most regulation applications use active filters for reactive and active power compensation. The work in [39] shows that the use of filters can be a solution for reactive power compensation, however the work in [40] shows that this method is not robust due to the harmonic distortion problems[41] that remain present in the system

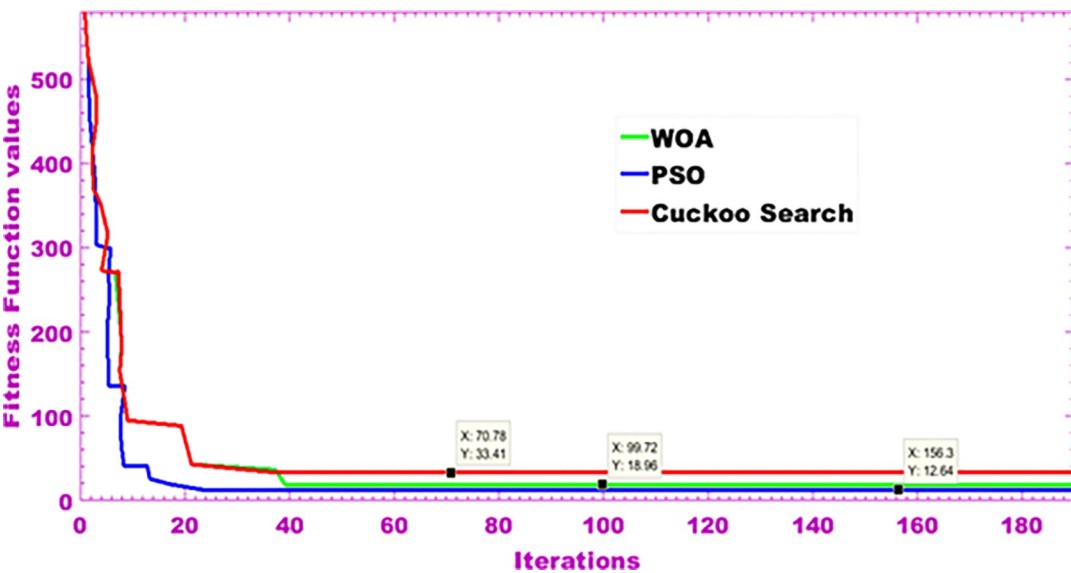

**Fig 13. Fitness function convergence with MPPT methods.**

when distributed generations are injected into the power networks. The use of these algorithms in this work offers the possibility of determining the size of the proposed system as well as the appropriate positions for the injection of renewable energy sources.

## 8. Conclusion

In this work, some algorithms have been used to solve the problem of power losses and abnormal voltage drops. We have techniques such as Whale Optimization Algorithms (WOA), PSO, GA, and cuckoo search algorithms. The aim of this study is to determine the optimal and appropriate location of renewable energy production units. This investigation led to the evaluation of the size of radial distribution systems on IEEE 69 and IEEE 33 buses. The search for an adequate stability margin for the power and voltage profile is highly dependent on the precise design and size of the distributed generation (DG) units at each node of a radial distribution system. The results obtained show a significant reduction in: power losses, voltage drop and line current losses. A comparative study of the selected algorithms shows the performance of the PSO algorithms over the other selected algorithms. Performance is shown in terms of power loss reduction and system behavior, which remains within the defined margins in terms of compliance with the operating constraints of the overall system. The superiority of the chosen method lies in the fact that, in the presence of harmonic disturbances and power losses, the system is automatically reconfigured by opening and closing certain system switches. Assessing the size of the overall system makes it possible to identify the appropriate positions where secondary sources can be injected in order to reinforce the system chosen for the study. Using this method, compared with recent work in the literature, it is possible to strengthen

**Table 3. *Comparison of the selected techniques.***

| Techniques | Fitness Values |
| --- | --- |
| WOA [42] | 18.96 |
| Cuckoo search [43] | 33.41 |
| PSO | 12.64 |

power systems in radial configuration and maintain system stability during transient periods or during the stochastic behavior of the system.

## Supporting information

**S1 File.**
(DOCX)

## Author Contributions

**Conceptualization:** Dieudonné Kidmo Kaoga, Kitmo.

**Data curation:** Kitmo.

**Formal analysis:** Bachirou Bogno, Deli Goron, Nisso Nicodem, S. Shanmugan, Dieudonné Kidmo Kaoga, Kitmo, Akhlaque Ahmad Khan, Yasser Fouad.

**Investigation:** Bachirou Bogno, Deli Goron, Nisso Nicodem, S. Shanmugan, Dieudonné Kidmo Kaoga, Akhlaque Ahmad Khan, Yasser Fouad.

**Methodology:** Bachirou Bogno, Deli Goron, Nisso Nicodem, S. Shanmugan, Dieudonné Kidmo Kaoga, Kitmo, Akhlaque Ahmad Khan, Yasser Fouad.

**Project administration:** Dieudonné Kidmo Kaoga.

**Resources:** Dieudonné Kidmo Kaoga.

**Supervision:** Dieudonné Kidmo Kaoga, Michel Aillerie.

**Validation:** Dieudonné Kidmo Kaoga, Michel Aillerie.

**Writing – original draft:** Dieudonné Kidmo Kaoga.

**Writing – review & editing:** Bachirou Bogno, Dieudonné Kidmo Kaoga.

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
