## [Decision Letter · Decision Letter 0]

13 Aug 2024

PONE-D-24-19219Enhancing the power quality in radial electrical systems using optimal sizing and selective allocation of distributed generationsPLOS ONE

Dear Dr. Kidmo Kaoga,

Thank you for submitting your manuscript to PLOS ONE. After careful consideration, we feel that it has merit but does not fully meet PLOS ONE’s publication criteria as it currently stands. Therefore, we invite you to submit a revised version of the manuscript that addresses the points raised during the review process.

We look forward to receiving your revised manuscript.

Kind regards,

Ch. Rami Reddy, Post Doc

Academic Editor

PLOS ONE

Journal Requirements:

Reviewers' comments:

Reviewer's Responses to Questions

**Comments to the Author**

1. Is the manuscript technically sound, and do the data support the conclusions?

Reviewer #1: Yes

Reviewer #2: Yes

2. Has the statistical analysis been performed appropriately and rigorously? 

Reviewer #1: Yes

Reviewer #2: Yes

3. Have the authors made all data underlying the findings in their manuscript fully available?

Reviewer #1: Yes

Reviewer #2: Yes

4. Is the manuscript presented in an intelligible fashion and written in standard English?

Reviewer #1: Yes

Reviewer #2: Yes

5. Review Comments to the Author

Reviewer #1: The work presented by the authors on distributed generations is interesting. I recommend publication of this work in Plos One if the following corrections are made

1. Figures 2 and 3: which switches are open or closed during DG injection?

2. The authors have not addressed system reconfiguration, which could be a major contribution to this work.

3. Figs. 5 and 6: we only see 2DG, whereas in Figs. 2 and 3 the authors presented 3DG.

4. A comparison table should be provided for the algorithms selected in Fig. 11.

Reviewer #2: Comments:

1) How does the integration of distributed generation (DGs) affect the voltage and power profiles in radial distribution systems? Explain by graph.

2) Can you explain how hybrid DG systems using radial DNs can help mitigate power blackouts? Explain the procedure.

3) How does the proposed optimization technique improve the scalability and stability of the proposed system in terms of locating stable nodes and managing voltage fluctuations?

4) Why author us PSO as a solution tool, why not some other metaheuristic method?

5) What are the key factors considered when designing and sizing DG units at each node for stability margins?

6) What are the benefits of assessing voltage, current, and harmonic distortion rates at various injection points of distributed generators before and after reconfiguring the structure?

7) How does the system reconfiguration approach ensure minimal power losses and maximum network power size in radially constructed networks?

6. PLOS authors have the option to publish the peer review history of their article (what does this mean?). If published, this will include your full peer review and any attached files.

Reviewer #1: No

Reviewer #2: No

---

## [Author Response · Author response to Decision Letter 0]

16 Sep 2024

Reviewer #1: 

The work presented by the authors on distributed generations is interesting. I recommend publication of this work in Plos One if the following corrections are made

1. Figures 2 and 3: which switches are open or closed during DG injection?

Based on the reviewer’s comments, we have added Table 1 which contain the state of switches.

2. The authors have not addressed system reconfiguration, which could be a major contribution to this work.

As recommended by the reviewer, we have addressed the reconfiguration of the system. Related results are provided in Figure 11 and 12.

3. Figs. 5 and 6: we only see 2DG, whereas in Figs. 2 and 3 the authors presented 3DG.

The sources are defined by the photovoltaic system for DG1, the wind plant for DG2 and the electricity grid for DG3 in this manuscript. 3DGs label was used to design the Grid’s source. The revised manuscript has been updated accordingly.

4. A comparison table should be provided for the algorithms selected in Fig. 11. 

As recommended by the reviewer, we have provided compared results in Table 3.

Reviewer #2: 

1) How does the integration of distributed generation (DGs) affect the voltage and power profiles in radial distribution systems? Explain by graph.

In Figure 7, without the DG (magenta colour), the voltage drop is high. With 1DG (blue colour) the voltage decrease improves. When 2DGs (green colour) and 3 DGs (black colour) are installed, the voltage profile improves and the system becomes stable. The same applies to Figure 9: with 1DG (magenta colour) or (blue colour), the power losses are very significant; when 2DGs (black colour) or (red colour) are injected, the power losses progressively decrease.

2) Can you explain how hybrid DG systems using radial DNs can help mitigate power blackouts? Explain the procedure.

Figure 11 shows the reconfiguration of the radial system on the IEEE 33 bus standard test, while Figure 12 shows the automatic reconfiguration of the system on the standard and 69 bus test. This reconfiguration reduces power losses and improves the voltage profile as the number of DGs increases. In this case, opening or closing the switches increases or decreases the active and reactive power on the nodes and branches of the radial system. Initially, some switches are open, while others are closed, both for the IEEE 33 bus standard configuration (Figure 2 and 11) and for the IEEE 69 bus standard configuration (Figure 3 and 12). Table 1 shows the switch positions. Opening or closing is managed by PSO algorithms, which take account of blackouts and sudden and unexpected changes in the behaviour of non-linear loads.

Three scenarios are presented: Grid+Wind (Scenario 1), Grid+PV (Scenario 2) and Grid+Wind+PV (Scenario 3). The maximum power level is reached with the Grid+Wind+PV scenario. In all three cases, the branches are connected and disconnected to create other meshes and loops

3) How does the proposed optimization technique improve the scalability and stability of the proposed system in terms of locating stable nodes and managing voltage fluctuations?

Scalability and stability are defined by Equations (2), (3), (4) and (5). These equations define the constraints on the performance of the system to hold the voltage level stable, in order to produce energy compensation for energy management at the common point of coupling. Algorithms are important to implement these criteria in the objective function in Equation (9). This condition enables the assessment of power losses and power quality on the different branches of the radial network.

4) Why author use PSO as a solution tool, why not some other metaheuristic method?

The PSO algorithms are compared to two other algorithms in Figure 13. The speed of convergence of the objective function shows that PSO converges towards the best scores compared with the other two algorithms. These results are in the same direction as those obtained in [33]. The aim of optimising the system is to reduce power losses and improve the voltage profile on the various branches and nodes of the radial system. The algorithms also make it possible to reduce the transient interval in order to limit power losses. The work in [34] shows that PSOs are suitable for reducing power loss and are better suited to the constraints associated with the fast response of an electrical system. Table 3 shows the results obtained by the algorithms selected from the literature. The highlighted manuscript has been revised and updated.

5) What are the key factors considered when designing and sizing DG units at each node for stability margins?

The key factors in DG sizing are the minimisation of power power losses and the enhancement of voltage profiles as defined in Table 2. This table summarises the different voltage and power ranges obtained after automatic reconfiguration of the system. It can be seen that by adding DGs the voltage profile improves and the power losses also decrease.

6) What are the benefits of assessing voltage, current, and harmonic distortion rates at various injection points of distributed generators before and after reconfiguring the structure?

The benefits of assessing voltage, current, and harmonic distortion rates at various injection points of distributed generators before and after reconfiguring the structure is to assess the quality of the energy produced and the impact of DGs on the grid. We can see that without a battery bank, the system is less stable and the voltage profiles are not good, whereas after integrating a battery bank and 2 DGs, the system performs better in terms of stability and power quality.

7) How does the system reconfiguration approach ensure minimal power losses and maximum network power size in radially constructed networks?

The system is reconfigured by opening some switches, as demonstrated in Figure 11 and Figure 12, and by closing the switches involved in switching. The system ensures automatic reconfiguration to reduce power losses and to meet the energy demand by the load at the common point of coupling. Energy management and system reconfiguration depend on the power required by the loads. In this case, opening or closing allows the system's active and reactive power to be increased or reduced.

Thank you very much for your helpful feedback.

---

## [Decision Letter · Decision Letter 1]

24 Oct 2024

PONE-D-24-19219R1Enhancing the power quality in radial electrical systems using optimal sizing and selective allocation of distributed generationsPLOS ONE

Dear Dr. Kidmo Kaoga,

Thank you for submitting your manuscript to PLOS ONE. After careful consideration, we feel that it has merit but does not fully meet PLOS ONE’s publication criteria as it currently stands. Therefore, we invite you to submit a revised version of the manuscript that addresses the points raised during the review process.

We look forward to receiving your revised manuscript.

Kind regards,

Ch. Rami Reddy, Post Doc

Academic Editor

PLOS ONE

Journal Requirements:

Additional Editor Comments:

This paper needs some minor revisions, before it can be considered for publication

1. The literature review may be improved with recent 2024 literature

2. Highlight the author contributions as bullet points, that may attract the readers after publication

3. There are many english language errors. Proof read with native speaker in power quality, preferebly a professor in electrical power domain

4. The discussion section may be improved further with recent literature and results

5. Explain every variables used in the equations, verify all once again

Thanks

Reviewers' comments:

Reviewer's Responses to Questions

**Comments to the Author**

1. If the authors have adequately addressed your comments raised in a previous round of review and you feel that this manuscript is now acceptable for publication, you may indicate that here to bypass the “Comments to the Author” section, enter your conflict of interest statement in the “Confidential to Editor” section, and submit your "Accept" recommendation.

Reviewer #1: All comments have been addressed

Reviewer #2: (No Response)

2. Is the manuscript technically sound, and do the data support the conclusions?

Reviewer #1: Partly

Reviewer #2: Yes

3. Has the statistical analysis been performed appropriately and rigorously? 

Reviewer #1: Yes

Reviewer #2: Yes

4. Have the authors made all data underlying the findings in their manuscript fully available?

Reviewer #1: Yes

Reviewer #2: Yes

5. Is the manuscript presented in an intelligible fashion and written in standard English?

Reviewer #1: Yes

Reviewer #2: Yes

6. Review Comments to the Author

Reviewer #1: The authors have made the effort to adequately address my concerns and expectations, I recommend the publication of this paper in Plos One

Reviewer #2: (No Response)

7. PLOS authors have the option to publish the peer review history of their article (what does this mean?). If published, this will include your full peer review and any attached files.

Reviewer #1: No

Reviewer #2: No

---

## [Author Response · Author response to Decision Letter 1]

30 Nov 2024

This paper needs some minor revisions, before it can be considered for publication

We thank the editor for this constructive comment.

1. The literature review may be improved with recent 2024 literature.

We thank the editor for this helpful suggestion.

Some recent works are cited in this regard, such as:

i. R. P. Kumar and G. Karthikeyan, “A multi-objective optimization solution for distributed generation energy management in microgrids with hybrid energy sources and battery storage system,” J. Energy Storage, vol. 75, p. 109702, Jan. 2024, doi: 10.1016/J.EST.2023.109702.

ii. B. Yang et al., “Modelling, applications, and evaluations of optimal sizing and placement of distributed generations: A critical state-of-the-art survey,” Int. J. Energy Res., vol. 45, no. 3, pp. 3615–3642, Mar. 2021, doi: 10.1002/ER.6104.

iii. B. N. Reddy et al., “Wind turbine with line-side PMSG FED DC-DC converter for voltage regulation,” PLoS One, vol. 19, no. 6, p. e0305272, Jun. 2024, doi: 10.1371/JOURNAL.PONE.0305272.

iv. K. R. Cheepati, S. B. Daram, C. Rami Reddy, T. Mariprasanth, B. Alamri, and M. Alqarni, “Analysis of Transient Stability through a Novel Algorithm with Optimization under Contingency Conditions,” Energies 2024, Vol. 17, Page 4404, vol. 17, no. 17, p. 4404, Sep. 2024, doi: 10.3390/EN17174404.

v. P. Rani, V. Parkash, and N. K. Sharma, “Technological aspects, utilization and impact on power system for distributed generation: A comprehensive survey,” Renew. Sustain. Energy Rev., vol. 192, p. 114257, Mar. 2024, doi: 10.1016/J.RSER.2023.114257.

The Manuscript has been entirely formatted and revised, adding many recent references.

2. Highlight the author contributions as bullet points, that may attract the readers after publication

We thank the editor for this helpful suggestion.

The contribution has been highlighted in the revised version of the manuscript.

3. There are many english language errors. Proof read with native speaker in power quality, preferebly a professor in electrical power domain

We thank the editor for this helpful suggestion.

We have improved the writing language in this manuscript by revising the whole document.

4. The discussion section may be improved further with recent literature and results

We thank the editor for this helpful suggestion.

The results obtained are compared with recent work in the literature, according to the editor's recommendations.

5. Explain every variable used in the equations, verify all once again

We thank the editor for this helpful suggestion.

We have clarified and gave the meaning of the variables used in the revised version of the manuscript.

Reviewer #1: 

Reviewer #1: All comments have been addressed

We thank the reviewer for this approval.

Thank you very much one again for your helpful feedback.

---

## [Editor Report · Decision Letter 2]

9 Dec 2024

Enhancing the power quality in radial electrical systems using optimal sizing and selective allocation of distributed generations

PONE-D-24-19219R2

Dear Dr. Kidmo Kaoga,

We’re pleased to inform you that your manuscript has been judged scientifically suitable for publication and will be formally accepted for publication once it meets all outstanding technical requirements.

Kind regards,

Ch. Rami Reddy, Post Doc

Academic Editor

PLOS ONE
---

## [Editor Report · Acceptance letter]

16 Dec 2024

PONE-D-24-19219R2 

PLOS ONE

Dear Dr. Kidmo Kaoga, 

I'm pleased to inform you that your manuscript has been deemed suitable for publication in PLOS ONE. Congratulations! Your manuscript is now being handed over to our production team.

Kind regards, 

on behalf of

Dr. Ch. Rami Reddy 

Academic Editor

PLOS ONE